# Druggable Transient Pockets in Protein Kinases

**DOI:** 10.3390/molecules26030651

**Published:** 2021-01-27

**Authors:** Koji Umezawa, Isao Kii

**Affiliations:** 1Department of Biomolecular Innovation, Institute for Biomedical Sciences, Shinshu University, 8304 Minami-Minowa, Kami-ina, Nagano 399-4598, Japan; koume@shinshu-u.ac.jp; 2Laboratory for Drug Target Research, Faculty & Graduate School of Agriculture, Shinshu University, 8304 Minami-Minowa, Kami-ina, Nagano 399-4598, Japan

**Keywords:** folding intermediate, cryptic binding site, protein kinase, thermodynamic equilibrium, DYRK1A, FINDY, chemical screening

## Abstract

Drug discovery using small molecule inhibitors is reaching a stalemate due to low selectivity, adverse off-target effects and inevitable failures in clinical trials. Conventional chemical screening methods may miss potent small molecules because of their use of simple but outdated kits composed of recombinant enzyme proteins. Non-canonical inhibitors targeting a hidden pocket in a protein have received considerable research attention. Kii and colleagues identified an inhibitor targeting a transient pocket in the kinase DYRK1A during its folding process and termed it FINDY. FINDY exhibits a unique inhibitory profile; that is, FINDY does not inhibit the fully folded form of DYRK1A, indicating that the FINDY-binding pocket is hidden in the folded form. This intriguing pocket opens during the folding process and then closes upon completion of folding. In this review, we discuss previously established kinase inhibitors and their inhibitory mechanisms in comparison with FINDY. We also compare the inhibitory mechanisms with the growing concept of “cryptic inhibitor-binding sites.” These sites are buried on the inhibitor-unbound surface but become apparent when the inhibitor is bound. In addition, an alternative method based on cell-free protein synthesis of protein kinases may allow the discovery of small molecules that occupy these mysterious binding sites. Transitional folding intermediates would become alternative targets in drug discovery, enabling the efficient development of potent kinase inhibitors.

## 1. Protein Kinases

Enzymatic phosphorylation of a protein substrate with adenosine triphosphate (ATP) was described in 1954 [1]. Thereafter, human genome sequencing revealed more than 500 protein kinases, which have been classified according to sequence homology-based clustering and designated the kinome [2,3].

Protein kinases are of therapeutic interest because aberrant kinase activity due to genetic and sporadic mutation, chromosomal translocation, and/or gene amplification can cause diseases such as cancer, diabetes, inflammation, and neurodegeneration [4,5]. The first clinical protein kinase inhibitor, fasudil (HA-1077), was approved in 1995 for cerebral vasospasm [6], and thereafter, many protein kinase inhibitors have been approved. The remarkable economic success of imatinib, approved in 2001 as a small molecule inhibitor of Abelson tyrosine kinase (ABL), which is dysregulated by chromosome translocation in chronic myelogenous leukemia, kindled drug research and development targeting the protein kinases involved in other cancers and diseases [7,8,9]. However, the U.S. Food and Drug Administration–approved inhibitors only target approximately 20 protein kinases in the human kinome [4,8]. Thus, the human kinome is still an attractive drug target with a significant research scope [4,10].

### 1.1. Structure of the Kinase Domain

The catalytic domain of protein kinases shares a conserved core consisting of two lobes: the N-lobe, featuring a β-sheet with five strands, and the C-lobe, mostly comprising α-helices and loops (Figure 1). A hinge region connects these lobes, forming an ATP-binding cleft, a highly conserved catalytic scaffold [11,12], between these two lobes. ATP and the substrate bind to the cleft and a nearby recognition site, respectively; the gamma phosphate of ATP is transferred onto a hydroxyl group of the substrate [13,14,15,16]; the resultant ADP is released from the cleft [17,18,19]; and ATP is re-supplied to the cleft to drive the catalytic cycle.

Protein kinases also possess a flexible and non-structured element (length: 20–30 amino acid residues) in the C-lobe, termed the activation loop. This loop begins with an invariant “DFG” motif (Asp-Phe-Gly) near the ATP-binding cleft (Figure 1) [20,21] and is in a transition from disordered to ordered upon activation. The carboxyl-terminal regions of this loop are designated the “P+1 loop” and the “APE” motif (Ala-Pro-Glu). The P+1 loop forms a pocket that binds to residues in the substrate peptide [22]. The conserved APE motif stabilizes the activation loop and the P+1 loop.

### 1.2. Nascent Kinases

Protein kinases are translated from mRNA as polypeptides, then folding into a thermodynamically stable state (Figure 2). Nascent kinases without any post-translational modifications generally take an inactive conformation, or latent conformation, in which ATP and the substrate cannot bind to the kinase with the typical affinity. The activation loop near the ATP-binding cleft blocks ATP and substrate binding. In the inactive conformation, the DFG motif is typically positioned outside of the active site in the cleft, exposed at the surface of the kinase. Thus, inactive states of kinases are also denoted as DFG-out (Figure 3).

Several studies have demonstrated that the N-lobe in the inactive kinase tends to be unfolded due to the structural flexibility of the activation loop and bound by HSP90/CDC37 chaperone machinery [23,24,25]. Because inactive states are not crucial for the catalytic cycle, kinases probably exhibit various inactive conformations; however, a consensus nomenclature for the inactive states is lacking [21].

### 1.3. Phosphorylation-Induced Activation of Kinases

Phosphorylation of discrete serine, threonine, or tyrosine residues in the activation loop converts the inactive DFG-out conformation to the active state, denoted as DFG-in (Figure 3) [21]. The negatively charged phosphorylated residues interact with positively charged residues adjacent to or in the activation loop, thereby decreasing loop flexibility. The structural stabilization of the loop opens the ATP-binding pocket by repositioning and fixing the DFG motif inside the cleft (DFG-in), thus maintaining a catalytically competent state [26]. In the active state (Figure 3), the Asp of the DFG motif binds to a magnesium ion that interacts directly with the phosphate in ATP.

The phosphorylation of specific residues in the activation loop occurs through various intermolecular or intramolecular mechanisms (Figure 4). In intermolecular mechanisms, an upstream active kinase phosphorylates a specific residue in the activation loop of a downstream kinase in a cascade. Mitogen-activated protein kinases (MAP kinases) belong to distinct signaling cascades in which an upstream kinase phosphorylates and activates a downstream kinase in an intermolecular manner. Thus, a MAP kinase kinase kinase(s) phosphorylates and activates a MAP kinases kinase(s) that in turn phosphorylates and activates a MAP kinase (e.g., Raf-MEK-ERK1/ERK2) [27]. In the other mechanism, transmembrane receptor-type tyrosine kinases, such as growth factor receptors (for example, EGF or FGF receptors), phosphorylate each other intermolecularly via activation loop swapping, in which the growth factor ligand-induced dimerization of extracellular regions positions their cytoplasmic kinase domains in a face-to-face orientation (also termed *trans*-autophosphorylation) [28,29]. These two intermolecular mechanisms require direct contact between the activation loops of the inactive kinase and the active kinase (Figure 4).

Several kinases phosphorylate their own activation loops intramolecularly (also termed *cis*-autophosphorylation) (Figure 4) [26]. Intramolecular autophosphorylation is often observed in constitutively active kinases [30]; however, the mechanism behind this type of autophosphorylation of the activation loop remains elusive [16,26]. The fundamental question is how the inactive conformation catches ATP and transfers the phosphate group to the specific residue in its own activation loop. Paradoxically, the inactive state of the kinase catalyzes intramolecular autophosphorylation, indicating a unique intramolecular autophosphorylation-prone state that is structurally distinct from the active conformation. Although it has not been completely elucidated, the autophosphorylation-prone conformation in the inactive state has been explained by the thermodynamic equilibrium between the inactive and active states [26]. The kinase domain structure can fluctuate between the active and inactive states without activation loop phosphorylation [21], although the equilibrium is inclined mostly toward the inactive conformation in the absence of phosphorylation. Once the equilibrium has shifted to a putative autophosphorylation-prone state through environmental or allosteric changes that are not completely known, the kinase cleft spontaneously catches ATP and catalyzes the intramolecular autophosphorylation of its own activation loop.

On the other hand, kinases such as CaMKII and DAPK1 do not contain a phosphorylatable residue in the activation loop, indicating that the kinome contains several kinases that do not require activation loop phosphorylation for the proper folding of the ATP-binding cleft and optimal kinase activity [31].

## 2. Kinase Inhibitors Type I, II, and III

Almost all kinase inhibitors bind to the ATP-binding cleft, a druggable pocket, and compete with an abundant supply of cytoplasmic ATP (2–8 millimolar) [32,33]. ATP-competitive kinase inhibitors are categorized by the DFG motif’s orientation in the inhibitor-bound form [34]. Type I inhibitors typically occupy the ATP pocket of the active DFG-in conformation and bind to adjacent regions to achieve selectivity for their target kinases (Figure 5). However, designing a highly selective type I inhibitor was difficult because the ATP pocket is conserved between kinases, and type I inhibitors utilized common residues to bind to the ATP pocket.

To overcome the inadequate selectivity of type I inhibitors, type II inhibitors have been developed. One of the representative type II inhibitors, imatinib, inhibits ABL [35,36]. In addition to the ATP pocket, type II inhibitors occupy allosteric pockets (Figure 5) [37,38,39]. The allosteric pocket for the binding of type II inhibitors is present in the region where the DFG motif lies in the active conformation. Therefore, the allosteric pocket is closed in the active DFG-in state but open in the inactive DFG-out state. Type II inhibitors maintain the inactive state by transitioning the DFG motif into the “out” state or by inducing a dynamic displacement of the DFG motif in the active state by forcing a shift in the equilibrium between the active/inactive states. Therefore, thermodynamic equilibrium enables type II inhibitors to suppress active DFG-in kinase proteins with a phosphorylated activation loop. The allosteric pockets in the DFG-out motif region recognized by type II inhibitors are less-conserved between kinases. Thus, type II inhibitors generally exhibit higher selectivity against kinases than type I inhibitors, but there are important exceptions to this trend [40]. It has been demonstrated that several type II inhibitors exhibit poor selectivity, whereas a number of type I inhibitors are quite selective; therefore, inhibitor type does not dictate selectivity [40]. Recently developed type I kinase inhibitors are quite selective in comparison with type II inhibitors [41]. These selective type I inhibitors interact with unique structural features that can help distinguish the target kinase from others.

Although both type I and type II inhibitors have been clinically successful, highly selective kinase inhibition remains challenging [37]. Type I and II inhibitors frequently exhibit adverse side effects due to off-target inhibition. In addition, genomic or sporadic mutations in amino acid residues comprising the ATP pocket induce resistance to ATP-competitive kinase inhibitors [42]. Thus, approaches targeting sites other than the ATP pocket in kinases have been investigated.

Other allosteric pockets in the inactive DFG-out conformation have been targeted in drug discovery [34]. The small molecule GNF-2 binds to the myristate-binding site in the C-lobe of ABL [43,44]. Type III inhibitors preferentially occupy only the allosteric pocket of their target kinase (Figure 5), binding outside the ATP pocket and blocking the transition of the DFG motif to the active state. The allosteric pocket varies structurally between members of the kinase families. Type III inhibitors are expected to exhibit higher selectivity than type I and II inhibitors, resulting in low adverse side effects and high success rates in clinical trials. Thus, type III kinase inhibitors have attracted increasing attention within the drug discovery field.

For the discovery of drugs targeting kinases, identifying type II and III inhibitors has remained quite difficult, and most clinically approved kinase inhibitors are type I [45]. Thus, alternative systematic and straightforward approaches for identifying type II and III inhibitors have been pursued.

### Databases for Kinome-Wide Inhibitory Profiles

Many kinase inhibitors are currently being evaluated in clinical trials, and their data are publicly available. A curated, annotated, and updated database, termed Protein Kinase Inhibitor Database (PKIDB), gathers information regarding protein kinase inhibitors that have already been approved as well as those that are currently in clinical trials [45]. In addition, published datasets on kinase inhibitors and their inhibitory profiling against kinases have been curated in a database provided by the International Centre for Kinase Profiling within the Medical Research Council (MRC) Protein Phosphorylation Unit at the University of Dundee (http://www.kinase-screen.mrc.ac.uk/kinase-inhibitors). Comprehensive data for the target landscapes of 243 clinically-tested kinase inhibitors are also available as an open-access resource, revealing previously unknown targets for established drugs and offering a perspective on the druggable kinome [46]. ProfKin was developed as a versatile and comprehensive web server for structure-based kinase selectivity profiling (https://www.researchsquare.com/article/rs-36477/v1). KIDFamMap provides a kinase-inhibitor-disease family map (http://gemdock.life.nctu.edu.tw/KIDFamMap/). KLIFs is a kinase database that dissects experimental structures of catalytic kinase domains and the way kinase inhibitors interact with them (https://klifs.net/) [47]. KinoMine is a web portal to search and extract chemical and biological kinase knowledge (http://kinomine.icoa.fr/). The KinaMetrix web resource allows researchers to investigate the relationship between kinase conformations and inhibitor space [48]. These examples illustrate the active development and release of web-based open resources.

Druggable pockets in kinases can also be explored using the web-based database Kinase Atlas, which summarizes all the druggable structures of particular kinases [49]. The inactive DFG-out states are included to provide alternative pockets in the computational prediction and modeling approaches utilized after in silico screening [50,51,52,53,54]. Thus, computational approaches have also been developed to expand our knowledge of druggable chemical spaces in protein kinases. These approaches must be continuously revised to keep up with improvements in computer performance.

## 3. Understanding Inhibitor Interactions with Proteins

Three major concepts explain the binding of an inhibitor to the surface pocket in its target protein: (1) lock-and-key model, (2) induced-fit, and (3) conformational selection [55,56,57,58]. In the lock-and-key model, geometric shapes on the surface of the target protein are an exact fit for the complementary structure of the small molecule inhibitor. However, this model has been considered insufficient to explain the strong and stable interactions between proteins and small molecules. To explain the strength of the interaction, the other two models were established. In the induced-fit model, changes in the conformation of the surface pocket occur after the small molecule binds, stabilizing the bound surface. Thus, the small molecule induces a structure on its target protein that is convenient for binding. In the conformational selection model, binding-competent and non-competent surface pockets exist in thermodynamic equilibrium, and binding of small molecules to binding-competent pockets makes the equilibrium shift to the bound state. Thus, small molecules select for and stabilize the bound conformation. Although induced-fit and conformational selection are different, as described above, these two models are coupled in the interaction, indicating a mixed mechanism [59,60,61]. Based on these models, scientists have discovered several small molecules that strongly suppress the function of their target proteins, with some molecules being developed into pharmaceutical drugs.

For these three concepts on protein-small molecule interactions, the binding pocket must be apparent on the protein surface, and the pocket structure itself probably determines the druggability of a protein. On the other hand, proteins with no apparent surface pocket have recently emerged as alternative drug discovery targets. Druggable pockets can be hidden in protein surfaces that have otherwise been considered undruggable.

## 4. Autophosphorylation of the Activation Loop during Folding

The aforementioned kinase inhibitors target mature kinases, including DFG-in and -out structures, which are fully folded thermodynamically stable structures. On the other hand, kinases also have immature structures, such as transitional, metastable, and partially unfolded intermediates during the folding process, which catalyze intramolecular autophosphorylation to complete the maturation process (Figure 6) [26,30]. Furthermore, the transitional folding intermediates are structurally distinct from the mature form and can also be selectively targeted by small molecule inhibitors (Figure 6) [62].

Dual specificity tyrosine phosphorylation-regulated kinase (DYRK) and glycogen synthase kinase 3β (GSK3β) family kinases activate themselves via the intramolecular autophosphorylation of tyrosine residues on their own activation loops [30]. Interestingly, purified DYRK proteins with dephosphorylated tyrosine could not catalyze tyrosine autophosphorylation but could still phosphorylate serine and threonine residues present on substrates [63]. Lochhead et al. explained this paradox through the existence of a transitional intermediate state occurring when the nascent polypeptide is still bound to ribosomes during protein translation and hypothesized that a conformational change after completion of translation and the one-off inceptive tyrosine autophosphorylation leads to this change in residue specificity [63].

On the other hand, Walte et al. showed that purified DYRK1A protein was still capable of tyrosine autophosphorylation [64]. According to them, this discrepancy between their result and that reported by Lochhead et al. was due to conformational fluctuations in DYRK1A, which can adopt catalytic conformation states leading to either intramolecular tyrosine autophosphorylation or intermolecular substrate phosphorylation in thermodynamic equilibrium. This equilibrium may be affected by the folding state of DYRK1A, post-translational modifications, and experimental conditions such as reaction temperature or ionic strength.

Lochhead et al. also showed that the biochemical properties of the kinase transitional intermediate were distinct from that of the mature state through interactions with two different kinase inhibitors, suggesting a unique ATP pocket in the putative autophosphorylation-prone conformation [63]. Tyrosine autophosphorylation by the transitional intermediate of *Drosophila* DYRK2 was inhibited by purvalanol A, but not by 4,5,6,7-tetrabromo-1*H*-benzotriazole (TBB), whereas both inhibited substrate phosphorylation. Purvalanol A and TBB are structurally distinct ATP-competitive inhibitors. Thus, Lochhead et al. concluded that the transitional intermediate and the mature form have structurally distinct ATP pockets and predicted the existence of an inhibitor selectively targeting the transitional intermediate [63].

By contrast, Walte et al. indicated that purvalanol A and TBB both inhibit tyrosine autophosphorylation and substrate phosphorylation of serine and threonine [64]. They also showed that all tested compounds inhibited substrate phosphorylation with higher potency than intramolecular autophosphorylation, indicating that substrate phosphorylation is more sensitive to kinase inhibitors than autophosphorylation [64]. They pointed out that performing an autophosphorylation assay under the same conditions as substrate phosphorylation is impossible because of the distinct environments around substrate residues on the activation loop during intramolecular autophosphorylation and substrate phosphorylation. Thus, the biochemical properties of DYRK family kinases that are distinct between intramolecular tyrosine autophosphorylation and intermolecular phosphorylation of serine and threonine on substrates remain to be elucidated.

### 4.1. Identification of A Folding Intermediate–Selective Inhibitor of DYRK1A

The existence of an inhibitor selective for a kinase transitional intermediate, proposed by Lochhead et al. seemed an enticing possibility yet was difficult to investigate [63,65]. DYRK family kinases catalyze intramolecular autophosphorylation in their activation loop and autoactivate themselves [63]. This autophosphorylation occurs not only in *Escherichia coli* cells but also in a cell-free protein synthesis system with purified ribosomes and other factors for transcription and translation [63,64]. These results indicate that no other upstream enzymes are involved in autoactivation. Regarding the inactive state of kinases, Kii and colleagues predicted that for constitutively active kinases such as the DYRK family, there would be an inactive state, similar to a type II or III inhibitor-bound conformation, as one of the transitional intermediates before full activation after protein synthesis (Figure 6). In addition to the prediction by Lochhead et al. this idea encouraged us to find a transitional intermediate–selective inhibitor of DYRK1A [62].

These types of inhibitors may be missed in conventional screening methods that use a recombinant mature kinase. Kii and colleagues developed a cell-based assay and examined substrate TAU phosphorylation by DYRK1A exogenously expressed as a tet-on inducible system in the presence or absence of small molecules from a small-scale structurally focused library [62]. The small molecules were added to the cells before or after the induction of DYRK1A expression. If a small molecule selectively inhibited a transitional intermediate during the maturation process, substrate phosphorylation would be inhibited when the molecule was added before DYRK1A induction, but not when added after induction. According to this criterion for selectivity, we found such an inhibitor and named it folding intermediate–selective inhibitor of DYRK1A (FINDY) [62].

FINDY is a structural derivative of an ATP-competitive inhibitor RD0392 that acts against mature DYRK1A [62]. RD0392 was found in a conventional kinase assay by using recombinant DYRK1A protein purified from *Escherichia coli* cells. FINDY also competes with ATP in the DYRK1A pocket before DYRK1A maturation is completed. Treating DYRK1A-expressing cells with FINDY caused misfolding and subsequent DYRK1A degradation. Furthermore, FINDY inhibited the intramolecular autophosphorylation of the serine residue at position 97 of DYRK1A in cell-free protein synthesis, demonstrating that FINDY directly affects DYRK1A autophosphorylation. Cell-free protein synthesis is a quite promising and straightforward method to directly evaluate the co-translational protein folding process affected by small molecules. Moreover, FINDY distinguished DYRK1A and DYRK1B in a cell-based assay and in in vivo Xenopus embryogenesis, exhibiting its relatively high selectivity against DYRK1A. Thus, FINDY selectively and directly inhibits the DYRK1A folding process.

Identification of FINDY supports the existence of a transitional folding intermediate structurally distinct from the mature DYRK1A [62]. Folding intermediates are thermodynamically unstable compared with the fully folded structures, which indicates that the structure of folding intermediates fluctuates. The thermodynamically fluctuating structures can be bound by a small molecule and transitioned into a metastable complex or distorted inappropriately, leading to misfolding and a non-functional conformation.

The DYRK1A folding intermediate misfolded by FINDY was degraded by the proteasome in the cytoplasm [62]. However, misfolded proteins can form aggregates or amyloid fibrils. Thus, it should be noted that its cytotoxicity and degradability affect the usefulness of the folding inhibition and that the small molecule-mediated folding inhibition potentially induces unfolded protein responses.

### 4.2. Relationship between Folding Intermediates and the Inactive DFG-Out State

The selective inhibition of folding intermediates partially involves the actions of type II and III kinase inhibitors. These inhibitors target the inactive DFG-out state of protein kinases. In addition to folded kinases with an unphosphorylated activation loop, an inactive DFG-out-like state should exist just before the translated, and folded kinases bind ATP and catalyze intramolecular autophosphorylation.

Whether the FINDY-targeted folding intermediate is identical to an inactive DFG-out state of DYRK1A remains elusive. X-ray crystal structures of the mature DYRK1A complexed with canonical kinase inhibitors have been studied and deposited in the database PDB; however, we have no structural information on the transitional folding intermediate of DYRK1A or the inactive DFG-out form. This structural information would confirm an alternative druggable pocket on a protein kinase. The next challenge is to explore the structure of DYRK1A complexed with FINDY and other intermediate-selective inhibitors.

## 5. Targeting Mysterious Pockets

The transitional folding intermediate of a protein kinase represents an alternative target in drug discovery because targeting the kinase folding process can help identify inhibitors, including not only folding intermediate–selective but also type II or III inhibitors. The typical binding pockets for type II and III inhibitors are certainly open in a transitional folding intermediate because the transient pockets exist only for a limited duration. Although the lifetime of transient pockets is short, i.e., on the order of milliseconds during protein folding [66,67], FINDY was shown to bind to these pockets during cell-free protein synthesis, suggesting that these transient pockets, which are apparent only during the folding process, are druggable.

In drug discovery targeting these metastable pockets, the pockets are termed “cryptic binding sites. “This concept could correspond well with the principles of the development of type II and III kinase inhibitors, as described below. On the other hand, this concept of cryptic binding sites could also accelerate drug development targeting transient pockets during the transitional folding intermediate state because computationally predicted cryptic sites are considered to open with a certain probability during the folding process, also described below.

### 5.1. Cryptic Binding Sites

Cryptic binding sites are metastable pockets apparent on the protein surface only when a small molecule is bound (Figure 7) [68,69]. These sites are not easily detectable on the unbound structure. The protein structure is in thermodynamic equilibrium and fluctuates dynamically. In other words, the protein surface can take on various structures according to its thermodynamic equilibrium under particular physiological conditions. In structural biology, the most abundant and stable population of a given protein structure is usually observed, while the rare and unstable structures are missed, although these can also be targeted by small molecules [70,71,72]. Because many proteins appear to lack druggable surface pockets, the concept of cryptic binding sites has received considerable attention, as this would expand the druggable proteome.

The inclusion of inactive DFG-out structures in cryptic binding sites may seem reasonable because inhibitor-binding pockets are not apparent on the active kinase. Type II and III inhibitors can inhibit target kinases with the activation loop phosphorylated because the active state contains potentially the inactive DFG-out structure with small population in its conformational ensemble. Thermodynamic equilibrium underlies the inhibitory mechanism of cryptic binding sites. Thus, the concept of cryptic binding sites has already been explored in drug discovery targeting protein kinases. However, even for drug discovery targeting the inactive DFG-out conformation of protein kinases, discovering cryptic binding sites and identifying inhibitors of such sites remains quite difficult.

Cryptic binding sites are metastable pockets in thermodynamic equilibrium, suggesting that the pockets open in a thermodynamically unstable state rather than in a static state. One of the thermodynamically unstable states in living cells is the transitional folding intermediate. Some of the thermodynamically fluctuating structures may bind a small molecule tightly during the folding process and subsequently transition into a misfolded state. This possibility is supported by the identification of FINDY. The structure targeted by FINDY has not yet been determined, but it must be structurally distinct from the known structure of DYRK1A. However, solving the structure of protein folding intermediates remains quite difficult.

### 5.2. Prediction of Cryptic Binding Sites

To identify cryptic binding sites in unbound protein structures, feasible computational simulations based on molecular dynamics (MD) simulations, and machine learning approaches have been developed [73,74,75]. The CryptoSite Web server is a tool that automatically predicts cryptic binding sites in protein structures [76,77]. With the help of CryptoSite, the size of the potentially druggable human proteome has increased from approximately 40% to approximately 78% of disease-associated proteins. However, predicting conformational changes after small molecule binding remains challenging. Continuous development of computational methodologies for delineating the thermodynamic structure of proteins would improve the predictions’ precision.

MD simulations have provided informative insights into binding sites and conformational changes. Some techniques using MD simulation have also been developed for detecting cryptic binding sites. Probe molecules are widely used in the simulation systems. Hydrophobic or charged molecules, such as benzene and acetate, are selected as probes. A multiple-probe method has recently been reported [78]. In that report, starting from the unbound structure, sampling the probe molecule densities after a 50-ns simulation mapped the cryptic binding sites. Although probe molecule methods are efficient at detecting cryptic binding sites, μs-order long-time simulations without probe molecules showed that the sampled conformations involved the solvent-exposed (opened) form at the cryptic binding site [79]. This means that the structure of the cryptic binding site is in fluctuation and opens transiently.

### 5.3. An Evaluation Method for Cryptic Inhibitor-Binding Sites

A method for validating the predicted cryptic sites is still a point of concern. The predicted cryptic binding sites in the target protein should be evaluated in biological assays, but evaluating these sites in in vitro assays using purified recombinant proteins may not be effective. Recombinant proteins are structurally stable, and the in vitro assay conditions are optimized to achieve full activity. Compared with in vitro assays using recombinant proteins in a test tube, cellular assays may be more suitable for the evaluation because the cytoplasmic molecular crowding conditions would diversify the structures of target proteins, some of which may then exhibit cryptic binding sites. However, cellular assays are troublesome and not suitable for the high-throughput screening of large-scale chemical libraries. Therefore, we suggest a validation strategy targeting the protein folding states without the use of living cells. Cryptic binding sites may open transiently with a decrease in the structural stability of the target protein, and during the folding process, a nascent protein fluctuates between folded and unfolded states.

A method to test proteins ranging from fully folded structures to thermodynamically unstable states during the folding process should be cell-based, but feasible drug discovery studies need a convenient method without living cells. We suggest a method based on cell-free protein synthesis followed by the measurement of enzymatic activity on the translated target protein, as this process could work well in any biological laboratory [62]. The transitional folding intermediates of the target protein certainly exist during the synthesis process. For protein kinases, the inactive DFG-out state targeted by type II and III inhibitors should also appear during the co-translational folding process. If the small molecule binds to and stabilizes the inactive state, the resulting translated kinase loses its enzymatic activity. This methodological concept has been supported by the study of Kii et al. [62], in which the kinase DYRK1A was translated using cell-free protein synthesis in the presence or absence of FINDY and then detected with an antibody against the specific autophosphorylation site. This method could be useful for evaluating not only folding intermediate–selective inhibitors but also type II and III inhibitors that target the inactive DFG-out state.

## 6. Cell-Free Protein Synthesis

Cell-free protein synthesis is an established method for producing recombinant proteins of interest in test tubes without living cells [80,81,82]. This system usually uses crude cell extracts prepared from chosen prokaryotic or eukaryotic cells [83]. Such cell extracts contain all components necessary for transcription and translation. Feasible and detailed protocols are currently available for the preparation [84]. In addition, a reconstituted cell-free protein synthesis system is available, in which all the components are added as factors that are produced and purified individually, leading to identical and consistent components [85]. The PURE (protein synthesis using recombinant elements) system is tightly controllable compared with crude cell extracts because component concentrations can be customized. A wide range of cell-free systems are commercially available; these systems are based on cell lysates derived from various sources or the PURE system. The cell-free protein synthesis system has continuously been improved [86].

Several concerns are associated with the synthesis of mammalian proteins, one of which is the availability of molecular chaperones. Protein kinases require a specific co-chaperone, CDC37, and the chaperone HSP90 to fold properly [23,24,87,88,89]. In addition, kinases can be classified by their dependency on the HSP90/CDC37 system as either a binary client or non-client [90]. In an immunoprecipitation experiment, DYRK1A did not interact with HSP90/CDC37, indicating that DYRK1A is a non-client kinase [88,90]. However, DYRK1A requires HSP90/CDC37 to fold appropriately and achieve full activity [88]. Furthermore, mutant DYRK1A mimicking the inactive state strongly interacted with HSP90/CDC37 [88]. These results demonstrate that the non-client kinase DYRK1A transiently interacts with the HSP90/CDC37 system during its folding process and dissociates upon completion of proper folding. This type of involvement of HSP90/CDC37 in the kinase folding process may apply to other kinases. The kinase activity of DYRK1A is low when it is synthesized without these chaperones.

On the other hand, the low kinase activity does not mean that molecular chaperones should necessarily be added. Assay throughput and dynamic range depend on the detection method. Therefore, the development and improvement of a cell-free detection method may be more worthwhile than chaperone addition. Enzymes for post-translational modifications can also be added to the cell-free protein synthesis system. If these extra enzymes are added into the cell-free system and if a small molecule is tested and results in a ‘hit’ as a positive inhibitor, it is necessary to rule out the possibility that the molecule inhibits the extra enzymes instead of the protein kinase.

Another concern is the translation speed of ribosomes. Appropriate translation speed underlies functional folding because translated peptides on ribosomes fold co-translationally [87,91]. Translation speed depends on tRNA supply, biased codon patterns, and ribosome function. For example, an excess supply of tRNAs could accelerate the translation speed, which results in faster polypeptide production on the ribosome than that under physiological conditions. If the speed of polypeptide production is beyond the capacity of co-translational folding, the translated polypeptide will be misfolded. Therefore, a decrease in the translation speed would promote proper folding and provide reliable enzymatic activity. Protein synthesis at low temperature, reduction of tRNAs, and other partial modifications may be useful in reducing the translation speed. Taken together, cell-free protein synthesis coupled with proper folding will provide a robust assay platform for drug discovery targeting cryptic binding sites.

## 7. Conclusions

Drug discovery is an incredibly costly challenge because many projects fail during late-stage clinical trials. The pharmaceutical industry has shifted its major drug discovery platforms from small molecules to biologics, such as antibodies and their drug conjugates, and gene and cell therapies. However, these biologics have a high cost of production and consequently are relatively unaffordable for patients. Compared with biologics, small molecules have several advantages, including a relatively simple chemical structure and oral availability, and they are cost-effective not only in terms of research and development but also for industrial-scale synthesis and quality management. Thus, innovative ideas have long been awaited in the small molecule-based drug discovery.

Cryptic inhibitor-binding sites expand druggable chemical spaces in the proteome. The type II and III kinase inhibitors that target cryptic binding sites in the inactive DFG-out state exhibit relatively high selectivity. Thus, small molecules targeting cryptic binding sites provide an opportunity to overcome failures in late-stage clinical trials without any adverse side effects. In-cell nuclear magnetic resonance (NMR) studies have demonstrated that the in-cell structures of proteins fluctuate to a greater extent than the structures in vitro because of molecular crowding effects or intracellular proteins [92], suggesting that cryptic binding sites may be more apparent in living cells. In vitro assays mimicking the cytoplasmic environment should be used during chemical screening targeting cryptic binding sites. In the near future, computational performance will reach a level capable of simulations covering the entire folding process, from the polypeptide to the protein tertiary structure and thermodynamic fluctuations in cellular macromolecular crowding conditions, allowing the rational design of highly selective inhibitors targeting cryptic binding sites.

## Figures and Tables

**Figure 1 molecules-26-00651-f001:**
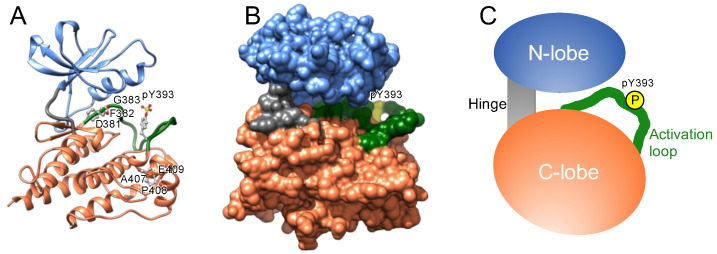
Structure of the kinase domain. (**A**) Ribbon representation of the kinase domain of ABL (Protein Data Bank (PDB) ID: 2V7A). Blue and orange regions indicate the N- and C-lobes, respectively. Gray and green loops represent the hinge region and activation loop, respectively. The tyrosine residue at position 393 in the activation loop is phosphorylated as indicated. The conserved amino acid residues (DFG and APE motifs) in and around the activation loop are also indicated. (**B**) Molecular surface image of the ABL domain (PDB ID: 2V7A), in which the colors are identical to the regions shown in (**A**). ATP binds to the cleft between the N- and C-lobes. (**C**) Graphical representation of the kinase domain.

**Figure 2 molecules-26-00651-f002:**
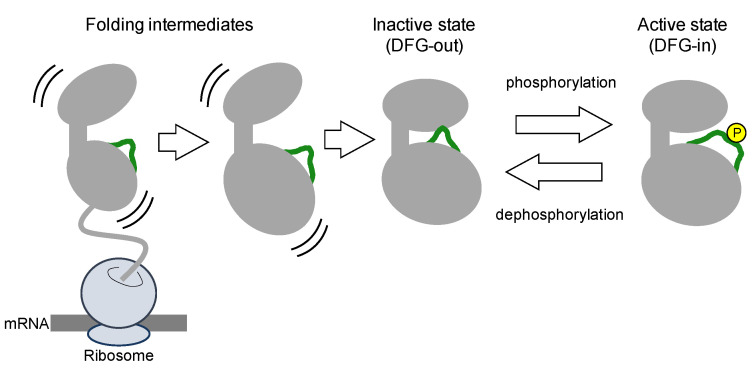
The co-translational folding process of the kinase domain and subsequent activation by phosphorylation (yellow) of the activation loop (green). The nascent polypeptide on the ribosome is folded into its intermediate state. Folding intermediates are unstable and exhibit structural fluctuations in thermodynamic equilibrium. Once the activation loop is phosphorylated, the intermediate state is converted to the active DFG-in state, which returns to the inactive DFG-out state after dephosphorylation by phosphatases.

**Figure 3 molecules-26-00651-f003:**
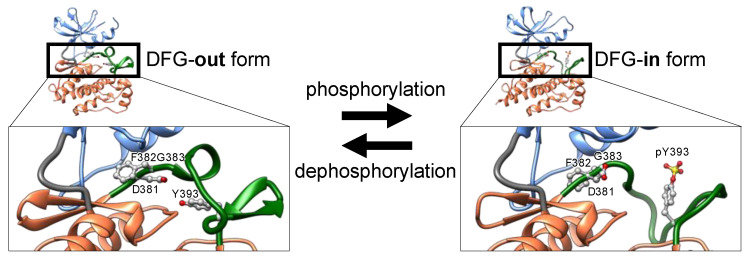
Enlarged view of the conformational transition of the ABL kinase domain between inactive DFG-out and active DFG-in states by activation loop phosphorylation and dephosphorylation. Structures of the DFG-out and -in forms are illustrated from PDB IDs 2HYY and 2V7A, respectively. The colors indicate the same regions shown in Figure 1. The orientation of the phenyl group of F382 is dynamically altered in the transition.

**Figure 4 molecules-26-00651-f004:**
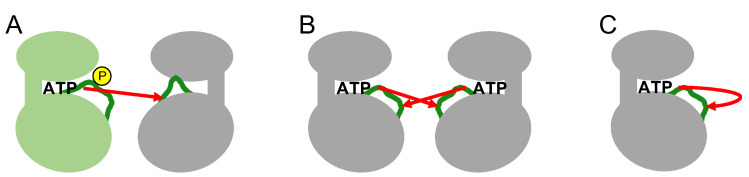
Graphical representation of activation loop phosphorylation. The green loops are the activation loops. The red arrows indicate the transfer of the phosphate group from ATP. (**A**) An active kinase with a phosphorylated (yellow) activation loop phosphorylates the activation loop of an inactive kinase. (**B**) Intermolecular autophosphorylation of activation loops of proximal kinases. (**C**) Intramolecular autophosphorylation of the activation loop of an inactive kinase.

**Figure 5 molecules-26-00651-f005:**
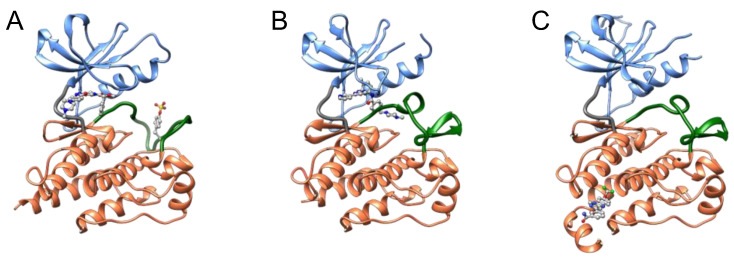
Ribbon representation of ABL kinase domains complexed with the indicated inhibitors. The inhibitors are presented as ball and stick models. The colors indicate the same regions shown in Figure 1. Complexes with the type I kinase inhibitor PHA-739358 (**A**), type II inhibitor imatinib (**B**), and type III inhibitor GNF-2 (**C**) are illustrated from PDB IDs 2V7A, 2HYY, and 3K5V, respectively. PHA-739358 and imatinib bind to the ATP-binding cleft of the active DFG-in state and the inactive DFG-out state, respectively. Imatinib binds behind the activation loop (green) in the ATP pocket (**B**). GNF-2 binds to the allosteric site in the C-lobe (orange).

**Figure 6 molecules-26-00651-f006:**
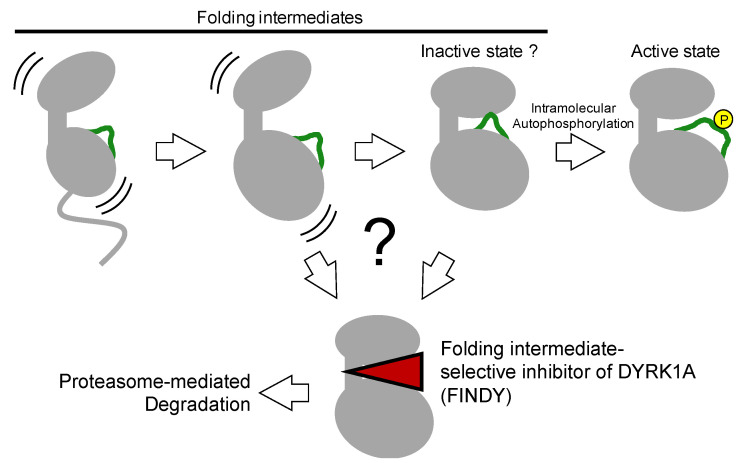
Graphical representation of the folding intermediate selective inhibition of the protein kinase DYRK1A by FINDY. FINDY selectively binds to the folding intermediate and inhibits the progression of the folding process, resulting in proteasome-mediated degradation in the cytoplasm. DYRK1A autophosphorylation has been demonstrated to be a one-off inceptive event required for activation [62,63]. Once the folding process is completed, DYRK1A cannot re-autophosphorylate. On the other hand, FINDY does not inhibit the fully folded form of DYRK1A, which corresponds to the active state. The intermediate state targeted by FINDY has not been clarified.

**Figure 7 molecules-26-00651-f007:**
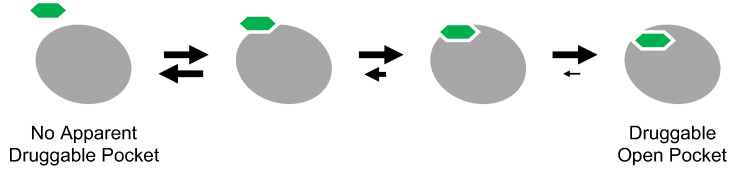
Graphical representation of cryptic binding sites. A small molecule (green) approaches the protein surface (gray). Interactions between the small molecule and the protein alter the surface structure (induced-fit), a binding-competent surface conformation of the thermodynamically fluctuating protein structure is caught by the small molecule (conformational selection), or a mixed mechanism underlies the interaction. Small molecule binding consequently reveals a hidden pocket on the protein surface.

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
