# Peer review of "Druggable Transient Pockets in Protein Kinases"

_molecules, 2021, doi:10.3390/molecules26030651_

Round 1

Reviewer 1 Report

The manuscript by Umezawa and Kii presents a review of literature on development of efficient and selective protein kinase inhibitors. The authors outline main problems in finding inhibitors that act selectively on a certain kinase on a small group of kinases. For this purpose, they describe total "topology" of kinase structure and present classification of small molecules into possible Type I, II, and III inhibitors depending on the binding site location and a kinase phosphorylation state. This part of the manuscript seems very helpful for understanding the main idea of the review focusing on finding transient pockets and cryptic binding sites for effective drug discovery strategy which leads to highly selective kinase inhibitors with a higher probability of success in further clinical trials. Efforts of different groups of authors working in this area are reviewed in a systematic and unbiased manner. The cited papers include both very recent works and key representative articles published in previous decades. The approaches described and reviewed in the manucsript can be very fruitful in discovery of compounds interacting with other types of biotargets, not only with protein kinases.

In my opinion, the review is important and well-written. It will be very interesting to the journal readership. I suggest acceptance of the manuscript for publication in the present form.

Author Response

Response:

We thank Reviewer #1 for the very positive evaluation of our manuscript and appreciate Reviewer #1 for the thoughtful review.

Reviewer 2 Report

Koji Umezawa and Isao Kii present a remarkable article entitled "Druggable transient pockets in protein kinases". The article is very well written and well organised, and I have to congratulate them. It is rare nowadays to review such high-quality articles.

While I heard about FINDY, a DYRK1A inhibitor, I discover in this article that it is presented as a folding intermediate-selective inhibitor of DYRK1A, an interesting metastable conformation which still has to show its value in drug discovery. As mentioned by the authors, unfortunately the crystal structure of FINDY-DYRK1A complex is not yet available preventing a clear understanding of the impact of the binding of FINDY in the structure of DYRK1A.

There are few sentences that should be revised:

Line 152 and line 173: It was true in ~2010 that Type I and Type II inhibitors were expected to be less and more selective respectively. However, this is not the case today, most of the Type I inhibitors developed nowadays are highly selective. In addition, there are many Type II inhibitors which are not selective.

Line 392: It is mentioned that Type II can bind to the DFG-in state of the protein, but based on the structure of Type-II inhibitor, the extended moiety which bind into the allosteric back pocket would have a steric hindrance with the phenylalanine of the DFG. It is not clear to me how a Type II inhibitor can bin in the DFG-in pocket.

Line 390: it is written that the pockets targeted by Type II inhibitors appear only when the compound is bound. This sentence is incorrect, since there are many apo structures of inactive conformation in DFG-out state crystallised.

Based on the two statement above, the conclusion should be modified.

I would suggest the authors to add information of KLIFs (https://klifs.net) and Kinomine (http://kinomine.icoa.fr) databases.

Author Response

Response:

We thank Reviewer #2 for the very positive evaluation of our manuscript and appreciate Reviewer #2 for the appropriate comments. We hope that our revised manuscript is satisfactory to Reviewer #2.

Comment No. 1:

Line 152 and line 173: It was true in ~2010 that Type I and Type II inhibitors were expected to be less and more selective respectively. However, this is not the case today, most of the Type I inhibitors developed nowadays are highly selective. In addition, there are many Type II inhibitors which are not selective.

Response No. 1:

We sincerely appreciate Reviewer #3 for the critical comment. We newly cited two papers (ref. No. 40 and No. 41) to support Reviewer #3's comment, and rephrased the sentence on page 4, lines 154-156 in the revised manuscript, and added the following sentences on page 5, lines 181-187 in the revised manuscript.

"designing a highly selective type I inhibitor was difficult because the ATP pocket is conserved between kinases, and type I inhibitors utilized common residues to bind to the ATP pocket" (On page 4, lines 154-156 in the revised manuscript)

"type II inhibitors generally exhibited higher selectivity against kinases than type I inhibitors, but there are important exceptions to this trend [40]. It has been demonstrated that several type II inhibitors exhibit poor selectivity, whereas a number of type I inhibitors are quite selective; therefore, inhibitor type does not dictate selectivity [40]. Recently developed type I kinase inhibitors are quite selective in comparison with type II inhibitors [41]. These selective type I inhibitors interact with unique structural features that can help distinguish the target kinase from others." (On page 5, lines 181-187 in the revised manuscript)

Newly cited references

No. 40

Davis, M.I.; Hunt, J.P.; Herrgard, S.; Ciceri, P.; Wodicka, L.M.; Pallares, G.; Hocker, M.; Treiber, D.K.; Zarrinkar, P.P. Comprehensive analysis of kinase inhibitor selectivity. Nat Biotechnol 2011, 29, 1046-1051, doi:10.1038/nbt.1990.

No. 41

Zhao, Z.; Bourne, P.E. Overview of Current Type I/II Kinase Inhibitors. In Next Generation Kinase Inhibitors, P., S., Ed. Springer, Cham: 2020; pp. 13-28.

Comment No. 2:

Line 392: It is mentioned that Type II can bind to the DFG-in state of the protein, but based on the structure of Type-II inhibitor, the extended moiety which bind into the allosteric back pocket would have a steric hindrance with the phenylalanine of the DFG. It is not clear to me how a Type II inhibitor can bin in the DFG-in pocket.

Response No. 2:

We are afraid that the sentence in the original version would mislead Reviewer #3. As mentioned by Reviewer #3, a type II inhibitor cannot bind to the DFG-in pocket. Herein, we would like to say that a type II inhibitor can bind to the active state with the activation loop phosphorylated, not to the DFG-in state. A kinase with the activation loop phosphorylated spontaneously takes the active DFG-in state. However, the state is not fixed in the DFG-in and fluctuated between the DFG-in and -out states. Thus, we rewrote the sentence in the revised version as described below.

"Type II and III inhibitors can inhibit target kinases with the activation loop phosphorylated because the active state contains potentially the inactive DFG-out structure with small population in its conformational ensemble" (On page 10, lines 413-416)

Comment No. 3:

Line 390: it is written that the pockets targeted by Type II inhibitors appear only when the compound is bound. This sentence is incorrect, since there are many apo structures of inactive conformation in DFG-out state crystallised.

Response No. 3:

We thank Reviewer #3 for pointing out the incorrect sentence. We remove this sentence on page 10, line 413 in the revised version. The removal of this sentence does not affect the logic of the paragraph.

         The removed sentence is "The pockets targeted by type II and III kinase inhibitors appear only when these inhibitors bind."

Comment No. 4:

Based on the two statement above, the conclusion should be modified.

Response No. 4:

According to Reviewer #3's comment, we rewrote the sentence on page 12, lines 545-546 in the revised manuscript. Reviewer #3 has a fundamental concern about the comparison between type I and type II inhibitors and suggests that type I inhibitor is not non-selective. Thus, we remove the part on the comparison "than type I inhibitors targeting the ATP pocket in the active DFG-in state" from the indicated sentence, as below. By removing this part, the conclusion does not contain the statement on the comparison between type I and type II inhibitors.

"The type II and III kinase inhibitors that target cryptic binding sites in the inactive DFG-out state exhibit relatively high selectivity" (On page 12, lines 545-546 in the revised manuscript)

Comment No. 5:

I would suggest the authors to add information of KLIFs (https://klifs.net) and Kinomine (http://kinomine.icoa.fr) databases.

Response No. 5:

We thank Reviewer #3 for suggesting the important databases. We added these two databases in the revised manuscript, as described below.

"KLIFs is a kinase database that dissects experimental structures of catalytic kinase domains and the way kinase inhibitors interact with them (https://klifs.net/) [47]. KinoMine is a web portal to search and extract chemical and biological kinase knowledge (http://kinomine.icoa.fr/)." (On page 6, lines 222-225 in the revised manuscript)

Reviewer 3 Report

This a well-written review on the design and development of innovative classes of protein kinase inhibitors in recent times in the quest to obtain agents that show improved specificity for a target kinase for therapeutic purposes. The authors have done an excellent job at introducing each new concept, chemical and process, thus enabling a sensible and easily understandable flow of information to a reader throughout the manuscript. There is sufficiently-detailed evidence to support the authors’ arguments when required and the points are logically argued. The authors have also included links to important data bases and web-based tools that would be invaluable to researchers in the fields of structural biology and cell biology, and in others, including pharmacology. This reviewer has no major issues with the manuscript and would like to commend the authors for this informative piece of work. There are only 3 points that the authors could consider.

  1. Line 109-110. Please rephrase sentence. MAP kinases belong to distinct signalling cascades in which an upstream kinase phosphoryates and activates a downstream kinase in an intermolecular manner. Thus, a MAP kinase kinase kinase (s) phosphorylates and activates a MAP kinases kinase (s) that in turn phosphorylates and activates a MAP kinase (e.g. Raf-MEK-ERK1/ERK2).
  2. Fig 6. Please include a proteosome-mediated protein degradation pathway as described in the legend (line 253-4).
  3. Please comment on the possibility of getting an unfolded protein response in intact cells when protein folding is affected by agents that bind to sites during folding.

Author Response

Response:

We thank Reviewer #3 for the very positive evaluation of our manuscript and appreciate Reviewer #3 for the helpful comments. We hope that our revised manuscript is satisfactory to Reviewer #3.

Comment No. 1:

Line 109-110. Please rephrase sentence. MAP kinases belong to distinct signalling cascades in which an upstream kinase phosphorylates and activates a downstream kinase in an intermolecular manner. Thus, a MAP kinase kinase kinase (s) phosphorylates and activates a MAP kinases kinase (s) that in turn phosphorylates and activates a MAP kinase (e.g. Raf-MEK-ERK1/ERK2).

Response No. 1:

We thank Reviewer #2 for providing the alternative sentences for the detailed explanation of MAPK cascade. We replaced the sentences in lines 109-110 with the suggested ones, as described below.

"Mitogen-activated protein kinases (MAP kinases) belong to distinct signaling cascades in which an upstream kinase phosphorylates and activates a downstream kinase in an intermolecular manner. Thus, a MAP kinase kinase kinase(s) phosphorylates and activates a MAP kinases kinase(s) that in turn phosphorylates and activates a MAP kinase (e.g. Raf-MEK-ERK1/ERK2) [27]." (On pages 3-4, lines 110-114 in the revised manuscript)

Comment No. 2:

Fig 6. Please include a proteosome-mediated protein degradation pathway as described in the legend (line 253-4).

Response No. 2:

According to the Reviewer #2's suggestion, we added "Proteasome-mediated Degradation" in Fig. 6. Please see Fig. 6 as attached below.

Comment No. 3:

Please comment on the possibility of getting an unfolded protein response in intact cells when protein folding is affected by agents that bind to sites during folding.

Response No. 3:

According to the Reviewer #2's suggestion, we added the following sentences on page 9, lines 356-360 in the revised manuscript.

"The DYRK1A folding intermediate misfolded by FINDY was degraded by the proteasome in the cytoplasm [62]. However, misfolded proteins can form aggregates or amyloid fibrils. Thus, it should be noted that its cytotoxicity and degradability affect the usefulness of the folding inhibition and that the small molecule-mediated folding inhibition potentially induces unfolded protein responses." (On page 9, lines 358-362 in the revised manuscript)
